# Management and Outcomes in Anal Canal Adenocarcinomas—A Systematic Review

**DOI:** 10.3390/cancers14153738

**Published:** 2022-07-31

**Authors:** Vasilis Taliadoros, Henna Rafique, Shahnawaz Rasheed, Paris Tekkis, Christos Kontovounisios

**Affiliations:** 1Department of Surgery and Cancer, Imperial College London, London SW7 2BX, UK; v.taliadoros@smd18.qmul.ac.uk (V.T.); p.tekkis@imperial.ac.uk (P.T.); c.kontovounisios@imperial.ac.uk (C.K.); 2Department of General Surgery, The Royal Marsden Hospital, London SW3 6JJ, UK; shahnawaz.rasheed@rmh.nhs.uk; 3Department of General Surgery, Chelsea and Westminster Hospital, London SW10 9NH, UK

**Keywords:** anal canal adenocarcinoma, surgery, abdominoperineal excision of rectum (APER), chemotherapy, radiotherapy, chemoradiotherapy, overall survival, median survival, recurrence, local recurrence, distant metastases

## Abstract

**Simple Summary:**

Anal canal adenocarcinomas are a rare type of bowel cancer. For this reason, it is challenging to perform large studies in order to determine the optimal treatment strategy to achieve the best outcomes. Options for treatment include radiotherapy, chemotherapy and surgery. These treatments may be combined or used alone. Outcomes are regarded as survival after diagnosis and treatment, or the recurrence of the disease. There is no universal gold standard that exists, with wide variability in practice and therefore also in outcomes between institutions. Thus, by reviewing the body of literature on the subject matter, the hope is to establish a management algorithm that may be tested and refined going forward. This is the intention of this systematic review.

**Abstract:**

(1) Background: Anal canal adenocarcinomas constitute 1% of all gastrointestinal tract cancers. There is a current lack of consensus and NICE guidelines in the United Kingdom regarding the management of this disease. The overall objective was to perform a systematic review on the multitude of practice and subsequent outcomes in this group. (2) Methods: The MEDLINE, EMBASE, EMCARE and CINAHL databases were interrogated between 2011 to 2021. PRISMA guidelines were used to select relevant studies. The primary outcome measure was 5-year overall survival (OS). Secondary outcome measures included both local recurrences (LR) and distant metastases (DM). The Newcastle–Ottawa Scale (NOS) was used to assess the quality of studies retrieved. The study was registered on PROSPERO (338286). (3) Results: Fifteen studies were included. Overall, there were 11,967 participants who were demographically matched. There were 2090 subjects in the largest study and five subjects in the smallest study. Treatment modalities varied from neoadjuvant chemoradiotherapy (CRT), CRT and surgery (CRT + S), surgery then CRT (S + CRT) and surgery only (S). Five-year OS ranged from 30.2% to 91% across the literature. LR rates ranged from 22% to 29%; DM ranged from 6% to 60%. Study heterogeneity precluded meta-analysis. (4) Conclusions: Trimodality treatment with neoadjuvant chemoradiotherapy (CRT) followed by radical surgery of abdominoperineal excision of rectum (APER) appeared to be the most effective approach, giving the best survival outcomes according to the current data.

## 1. Introduction

Anal adenocarcinoma is a rare entity, making it hard for a consensus to be reached with regards to its management, as it is not possible conduct large-scale studies with a sufficient number of patients. Therefore, a retrospective approach is commonly used to study the benefits of potential treatments. There are currently no UK NICE guidelines on the management of anal adenocarcinoma. Reviewing the current literature and the data reported by previous studies will hopefully add more clarity to the effectiveness of the various treatment options or their combinations.

The variability in reported outcomes in the literature may be explained by the diagnostic dilemma of a rectal cancer versus anal cancer. To address this, The American Joint Committee on Cancer (AJCC) as well as the Union for International Cancer Control (UICC) have agreed on the definition of anal canal cancers to be tumours whose epicentres are situated between the anal verge and less than or equal to 2 cm above the dentate line. The anal canal has also been defined by the two bodies to span from the anorectal ring to the anal verge. Wider use of these definitions will reduce the heterogeneity of data in these groups [1,2].

There are many different types of anal cancer. The commonest form of anal cancer is squamous cell carcinoma (SCC) and makes up approximately 90% of all the cancers of the anal canal. The remaining 10% of anal cancers include anal adenocarcinomas, basal cell carcinomas, melanomas and other non-epidermoid cancers such as small cell and undifferentiated cancers, as well as lymphomas [1,2].

In the UK, between the years 2016 and 2018, there was an average of 1519 new cases per year. Each year, approximately 25% of all new cases in the UK were in people over 75 years of age. It is more common in females, and the rate of new cases in females has more than doubled recently, with an increase of around 117%. Further, there were 422 deaths in these years in the UK. The mortality rates are the highest in people over the age of 90, with 43% in people over the age of 75. Considering mortality rates of bowel and anal cancers combined, the rates are overall lower for people of non-White ethnicity compared to those for people of White ethnicity. In high deprivation areas, anal cancer deaths are more common [2].

In England alone, for the years 2013–2017, the 1-year survival rate for people diagnosed with anal adenocarcinoma was 84.8%, the 5-year survival rate was 58.7% and the 10-year survival rate was 52.2%. The survival rate drops from 99% if it diagnosed at its earliest stage to 53% if it is diagnosed at its latest stage, i.e., from almost everyone to one in two. It is thought that 91% of anal SCC cases in the UK are preventable, since 91% of cases are caused by infections, such as the Human Papilloma Virus (HPV) 16 and 18. However, the proportion of anal adenocarcinomas caused by HPV is not described. Other acknowledged aetiological factors for anal adenocarcinoma include smoking, age, sexual promiscuity and immune suppression. This is important to consider when thinking of public health measures and interventions to reduce rates [2].

A combination of medical and surgical treatment options is available for the treatment of anal adenocarcinoma. The jury is out as to the most effective treatment method. According to the National Comprehensive Cancer Network (NCCN) Clinical Practice Guidelines in Oncology, the management of anal adenocarcinoma consists of neoadjuvant therapy followed by APER, analogous to rectal cancer treatment [3].

This systematic review was conducted looking at anal adenocarcinomas, focusing on relevant literature found between 2011 and 2021, following on from a previous systematic review carried out by Anwar S et al. which included studies up to the year 2011 [4]. The rationale behind this review is to look at the latest literature and compare outcomes of different treatment paradigms. The lack of clear guidelines and the controversy as to the most effective treatment approach as outlined by different results makes further investigation of the latest literature a worthy pursuit. The hope is that the analysis of the latest data might shed more light about the efficacy of different measures and add to the existing body of evidence, and hopefully influence future guidelines.

## 2. Materials and Methods

### 2.1. Eligibility Criteria, Sources and Search Strategy

The PRISMA guidelines for systematic reviews were followed. The search was conducted using four databases, MEDLINE, EMBASE, EMCARE and CINAHL, using the HDAS platform. Inclusion criteria included any relevant research articles on the management of anal adenocarcinoma from 2011 to December 2021, in the English language with the selected search terms. The date range was selected to be from 2011 onwards in order to exclude another similar review that included articles up until that point. The keywords used in the search strategy can be found in Appendix A.

### 2.2. Selection and Data Collection Process

The search strategy followed is outlined in the PRISMA flowchart in Figure 1 below.

The initial search produced 3966 results, and since only one duplicate was found on HDAS, further deduplication was performed using ZOTERO. Subsequently, the results were manually scanned to remove any further duplicates that the automation process might have missed. A total of 1286 duplicate results were removed, leaving a final total of 2680. With the 2680 results, the list was manually reviewed to exclude any titles that were not specifically relevant to the management of anal adenocarcinoma, leaving a remainder of 80 article titles. One article was an abstract by Fet et al., of which the full text could not be found and thus was not retrieved, leaving 79 articles. Six additional articles that were deemed not relevant were removed, leaving 73 articles. Thirty-three articles with no abstract and 22 case reports were, excluded leaving 18 articles. Finally, three more titles that appeared to meet the selection criteria were removed, leaving a total of 15. One of them was a systematic review by Anwar S et al. [4] and another was a clinical review by Lukovic et al. [6]. The study by Ogawa et al. [7] had only two out of the total of 92 patients with Crohn’s-associated anal fistulae that were subsequently diagnosed as having adenocarcinomas, therefore, the study was excluded.

The Newcastle–Ottawa Scale (NOS) is a quality assessment scale used in past systematic reviews. A score was assigned to appraise the quality of each of the selected studies. All 15 remaining studies were deemed to be of sufficient quality, i.e., having a score of five or more, to be used in this review. The search identified studies providing treatment outcomes such as overall survival for patients with anal adenocarcinoma. The data were manually extracted and tabulated in Microsoft Excel. The total population size, number of patients with Crohn’s, median age values, treatment types, median follow-ups and outcomes such as recurrence and overall survival were tabulated.

## 3. Results

There was great variability in the data and management or treatment types in the selected studies. The treatment types of differences within and between the studies could be categorised as chemoradiotherapy alone (CRT only), surgery only (S only), surgery followed by postoperative/adjuvant CRT (S + CRT), neoadjuvant chemoradiation followed by surgery (CRT + S), radiotherapy alone (RT only) or chemotherapy alone (CT only). Table 1 outlines the OS values provided by the selected studies for each treatment type.

The studies by Bertelson et al. [9], Franklin et al. [10], Lewis et al. [15], Wang et al. [14], McKenna et al. [12], Leong et al. [13] and Gogna et al. [20] did not provide values for the 5-year overall survival rates or any OS rate values for each treatment group. In the Franklin et al. study of 307 patients, the 1-, 3-, 5- and 10-year survival rates were 76.8%, 46.2%, 30.2% and 16.2%, respectively. In addition, the study provided survival curves based on staging, and the median overall survival was 33 months.

The OS in the Su et al. retrospective study with 126 patients was 85.8% at 1 year, 62.5% at 3 years and 43.4% at 5 years post-treatment at follow-up [11]. No overall survival values by treatment or median overall survival were provided in this study. The median overall survival in the McKenna et al. study that examined 2117 patients was 65 months, and in the Leong et al. study of 5 patients it was 10.5 months. Using the online program Plot Digitizer, the data from the plots were extracted [23]. By extracting the data from the overall survival plot provided in the Lewis et al. study of 1183 patients, the 50-month, 100-month and 150-month OS values were found to be 61.6%, 40.9% and 27.0%. The 5-year OS was 55.9%, and the median overall survival was 72.5 months. In the 2019 study by Li et al. looking at 1747 patients [16], the 5-year OS was 61.1% for CRT + S and 39.8% for CRT only, and the median overall survival was 79 months for CRT + S and 42 months for CRT only.

The 2019 study by Malakhov et al. has shown a 48.4% total overall survival (OS) at a 5-year follow-up for anal adenocarcinoma [17]. The 5-year OS for different treatment groups was shown to be 57.6% for S only, 64.6% for CRT + S, 51.7% for S + CRT and 39.2% for CRT only. This study also compared survival rates between stage II and stage III of anal adenocarcinoma. Similarly, the 2019 study by Wegner et al. has estimated the 5-year OS to be 55% [18]. The 5-year overall survival rates by treatment type as outlined by the study’s plot were shown to be 69.1% for S only, 64.1% for CRT + S, 67.3% for S + CRT and 42.0% for CRT only. Their respective median survival rates were 78, 83, 92 and 46 months.

The 2020 study by Park has shown an overall cause-specific survival (CSS) rate of 72.9% at 3 years [19]. The 3-year CSS rates were 77.7% in the S-only group, 80.3% in the CRT + S group, 65.8% in the S + CRT group, 63.9% in the CRT-only group and, finally, 35.7% in the RT or CT-only group. In the 2020 Gogna et al. study the 1-, 3-, and 5-year OS rates were 76.1%, 52.6% and 39.6%, respectively. This study also provided survival curves based on different age groups and staging. The median overall survival in the Gogna study was given as a comparison between the patients receiving surgery and those that did not, with the former being 116.7 months and the latter 42.7 months.

### 3.1. Surgery Only (S)

The 2019 study by Malakhov et al. has shown the 5-year OS for different treatment groups to be 57.6% for the S-only group, whereas Wegner et al. have found an OS value of 69.1%. The study by Park has a 3-year CSS rate of 77.7% in the S-only group.

### 3.2. Surgery and Postoperative CRT (S + CRT)

The overall survival rate at 5 years in the Malakhov et al. study for S + CRT was 51.7%, with Wegner et al. showing a higher value of 67.3% for S + CRT. Park has shown a 3-year CSS rate of 65.8% in the S + CRT group.

### 3.3. Primary CRT (CRT)

Peiffert et al. [8], in their 2012 published phase III randomised clinical trial (RCT), compared the treatment of anal adenocarcinoma using chemoradiotherapy with or without induction chemotherapy. The study does not state overall survival values but provides the 3-year and 5-year colostomy-free survival (CFS) values. The CFS outcomes were 76% and 75% at 3 and 5 years, respectively, for CRT alone. For CRT with induction chemotherapy, the CFS values were 79% and 76.5% at 3 and 5 years, respectively. The overall survival at 5 years was 39.8% for the CRT-only group according to Li et al., almost identical to that from Malakhov et al. at 39.2%. Wegner et al. also reached a similar value of 42.0%. Park has shown a 3-year CSS rate of 63.9% in the CRT-only group. Median overall survival for the CRT-only group was found to be 42 and 46 months by Li and Wegner, respectively.

### 3.4. Neoadjuvant CRT and Surgery (CRT + S)

The 5-year OS in the studies by Li et al. and Malakhov et al. were once again very similar for CRT + S, 61.1% and 64.6%, respectively. Similarly, Wegner et al. have estimated the CRT + S group’s 5-year OS to be 64.1%. Park has shown a 3-year CSS rate of 80.3% in the CRT + S group. The study by Chatani et al. in 2021 [22] shows median overall survival of 85.8 months for the patient group receiving chemoradiation and local excision (LE), CRT + S (LE), as opposed to 65.3 months for the group receiving chemoradiation and abdominoperineal resection, CRT + S (APER). Median overall survival values for the CRT + S groups were found to be 79 months and 83 months by Li and Wegner, respectively.

### 3.5. Management of Anal Adenocarcinoma in Crohn’s Disease

In the study by Yasuhara et al. [21], the OS rates of anal adenocarcinoma were split into cases associated with Crohn’s disease (CA) and cases not associated with Crohn’s disease (N-CA), as well as into clinical tumour sizes (T1–2 vs. T3–4). For T1–2 tumours, the 5-year OS for CA was 91.0% compared to the N-CA, where it was 85.7%. For T3–4 tumours, the 5-year OS were 25.8% for CA cases compared to 71.0% N-CA cases.

The results of all the studies included in this review are outlined in Table 2 below.

## 4. Discussion

Anal adenocarcinoma is a relatively rare type of tumour compared to the more common squamous cell carcinoma of the anus. Due to its rarity and the lack of data, a general lack of consensus has existed so far with regards to its treatment and management. This has also been the case due to the heterogeneity of the groups found in different studies as well as the contrasting results by some of the studies. Studies conducted on anal adenocarcinoma have mostly been smaller retrospective ones and case reports or case series. Larger retrospective studies such as the one by Lewis et al. [15] provide a more accurate representation of the outcomes and efficacy or lack thereof of different treatments and their combinations.

Adenocarcinoma of the anus also exhibits more aggressive behaviour in comparison to that of the squamous cell type. This aggressive nature includes higher rates of local failure, distant metastasis and disease-associated mortality [9,24,25,26,27]. Low survival outcomes are also observed in the Franklin et al. and Lewis et al. studies [10,15]. More recent studies are also in agreement, such as the one by Wang et al. in 2019 that has shown adenocarcinoma of the anus having a worse prognosis than rectal adenocarcinoma and, in addition, stating that the T staging criteria for anal carcinoma may not be valid for staging anal adenocarcinoma [14].

The treatment approach usually taken for anal adenocarcinoma is that of rectal adenocarcinoma as opposed to the one usually taken for squamous cell carcinoma of the anus of definitive chemoradiation. Rectal adenocarcinoma is commonly treated using neoadjuvant chemoradiotherapy followed by surgical resection, which can be either local or radical. The mainstay of treatment in anal adenocarcinoma includes surgical resection, but there have been reports that CRT alone can also provide comparable survival outcomes [26,28]. It is worth noting though that Kounalakis et al. have shown a survival benefit in patients who undergo APER, regardless of whether radiation was part of the treatment.

Anal adenocarcinoma more frequently leads to distant metastases in comparison to other types of anal cancers such as squamous cell carcinomas, making the option of radical resection more limited [4,29]. In addition, APER is associated with significant morbidity and decrease in the quality of life of patients, making CRT a more favourable option for patients, since anorectal function is reserved [15,24]. The recent NCCN guidelines have tried to standardise the treatment of anal adenocarcinoma to tackle the lack of agreed-upon existing practice guidelines and based it on studies such as the one from Chang et al. in 2009 and Beal et al. in 2003 [25,30]. In the UK, however, there are currently no clear NICE guidelines pertaining to the management of adenocarcinoma of the anus.

### 4.1. General Interpretation of Results

Peiffert et al. have shown no advantage for the use of induction chemotherapy or a high-dose radiation boost. The 2019 study by McKenna et al. has found that patients receiving nonsurgical management have increased mortality, hence, the authors recommend a multidisciplinary evaluation and surgery. The study by Franklin et al. recommends a more aggressive approach due to worse prognosis for anal adenocarcinoma when compared to both rectal adenocarcinoma and squamous cell carcinoma. This seems to agree with the conclusions of Su et al. which advise on prophylactic inguinal node treatment for patients with anal adenocarcinoma, regardless of inguinal lymph node (ILN) status, i.e., even if ILNs are not positive.

Bertelson et al. have shown that curative resection provides no significant long-term outcomes when it comes to disease-free survival and recommend CRT followed by APER (CRT + S) as the treatment, at least as it relates to patients with stage II anal adenocarcinoma. Similarly, Leong et al. are also in favour of the multimodal treatment approach of neoadjuvant CRT followed by APER (CRT + S). Trimodality treatment refers to treatment with three modalities, i.e., chemotherapy, radiotherapy and surgical resection. This has also been shown by Lewis et al. to yield improved survival outcomes for patients, in particular CRT followed by APER within 6 months.

The main conclusions of each study selected in this systematic review have been summarised in Table 3, following the same structure as that from Anwar et al. for the year range 2011–2021 (see Appendix B Table A1).

Another clinical review by Lukovic et al. published in 2020 recommends a similar treatment approach to that of Lewis et al., i.e., trimodality therapy (CRT + S), with curative intent treatment for anal adenocarcinoma being neoadjuvant RT to the primary tumour, anal canal region and regional LNs and concurrent capecitabine or FU chemotherapy [6,11,26,29]. For worse prognosis cases with clinical T3/T4 or positive nodal metastasis, upfront chemotherapy followed by conventionally fractionated CRT can be an alternative approach. The recommendation for radical surgery such as APER in this study is between 3 and 6 months following termination CRT treatment [6,16,25].

CRT followed by surgery is associated with a significant benefit according to another study by Li et al. [16]. Furthermore, this conclusion is also echoed in the study by Wegner et al. that found improved overall survival rates for adenocarcinoma of the anus by incorporation of surgery when compared to that with CRT alone. Neoadjuvant CRT with APER was again found to be the treatment of choice for a more aggressive approach towards adenocarcinoma of the anus as per Malakhov et al. CRT given preoperatively with surgical resection was once again found to possibly maximise overall survival outcomes (CRT + S) according to Park in 2020. Surgery was also found to significantly improve overall survival outcomes in patients with anal adenocarcinoma as per Gogna et al.

An overview of the general management pathway for anal adenocarcinoma has been illustrated by the flowchart in Figure 2 below.

### 4.2. Limitations of Evidence and Review Processes Used

All studies are bound to be limited in some way. The Malakhov et al. study did not have any information on the type of chemotherapy or immunotherapy that was given to the patients or the exact cause of death, since this information was not available on the National Cancer Database (NCDB) used. The NCDB does not have information of the treatment decision making rationale, as noted by Lewis et al. when investigating the reasoning behind patients being offered APER or not. The NCDB does not include variables such as treatment toxicity, cancer-specific survival or local and regional control. In some of the studies, such as the Yasuhara et al. study, the limitation was the nature of the study itself, i.e., being a single-arm retrospective study done at a single institution. In the Bertelson study, one of the limitations was poor follow-up for some patients.

Selection bias is present in all retrospective studies that might lead to confounding. Sample sizes, treatment groups, population and tumour characteristics are also some other important differences to take into consideration when determining the outcomes of different treatment types. Limitations with these studies also include the fact that anal adenocarcinoma can sometimes be incorrectly classified at the diagnostic level into its close counterparts such as rectal adenocarcinoma and anal squamous cell carcinoma, thus having more cases omitted from retrospective studies.

### 4.3. Implications of Results for Practice, Policy, and Future Research

The rarity of anal adenocarcinoma should encourage more large retrospective studies to obtain more reliable data on the overall survival outcomes of patients depending on the different treatments they receive. For the results of further research studies to be more useful, a more homogeneous approach to data collection at the clinical level as well as a more standardised approach to the treatment comparisons need to be taken, which would make the results from different studies less heterogeneous and more appropriate for systematic reviewing and meta-analysing.

Of the 15 studies, eight have shown that multimodality or trimodality therapy, which includes chemoradiation (CRT) followed by surgery or abdominoperineal resection (APER), offers better survival outcomes than surgery (S) or CRT alone, including the ones by Leong et al. in 2019 [13], Lewis et al. in 2019 [15], Li et al. in 2019 [16], Malakhov et al. in 2019 [17], Wegner et al. in 2019 [18], Park in 2020 [19], and Gogna et al. in 2020 [20], and Yasuhara et al. in 2021 [21].

In addition, three out of the 15 studies have shown surgery to be beneficial in the management of anal adenocarcinoma, with the McKenna et al. 2019 [12] study showing that nonsurgical management is associated with increased mortality, and the Wegner et al. 2019 [18] and Gogna et al. 2020 [20] studies showing improved overall survival by incorporation of surgery in the management of anal adenocarcinoma. The Franklin et al. 2016 [10] study recommends more aggressive therapy, and the Su et al. recommends prophylactics of the inguinal lymph node.

## 5. Conclusions

Based on the current evidence collected from the studies included in this review, it is recommended that the trimodality treatment approach is followed as described by Lukovic et al. [6]. It has been demonstrated by many other variations of the same treatment combination of CRT followed by APER in the different retrospective study results included in this review that better survival outcomes are achieved. This recommendation is based on current evidence, and more research is encouraged to ensure that the treatment approach to anal adenocarcinoma is optimised and standardised. Furthermore, more exploration of the genetics of this type of tumour might open doors into new treatment strategies and pharmacological agents.

## Figures and Tables

**Figure 1 cancers-14-03738-f001:**
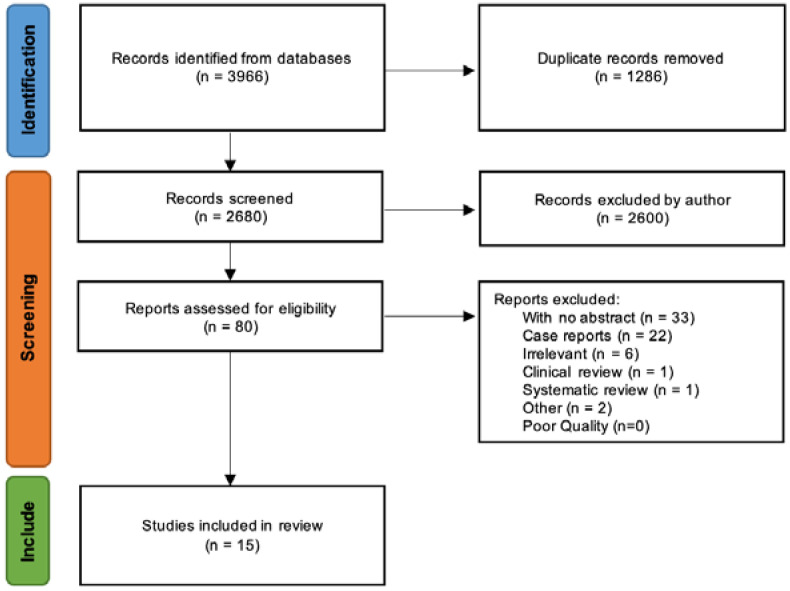
PRISMA flowchart outlining search strategy (PRISMA [5]).

**Figure 2 cancers-14-03738-f002:**
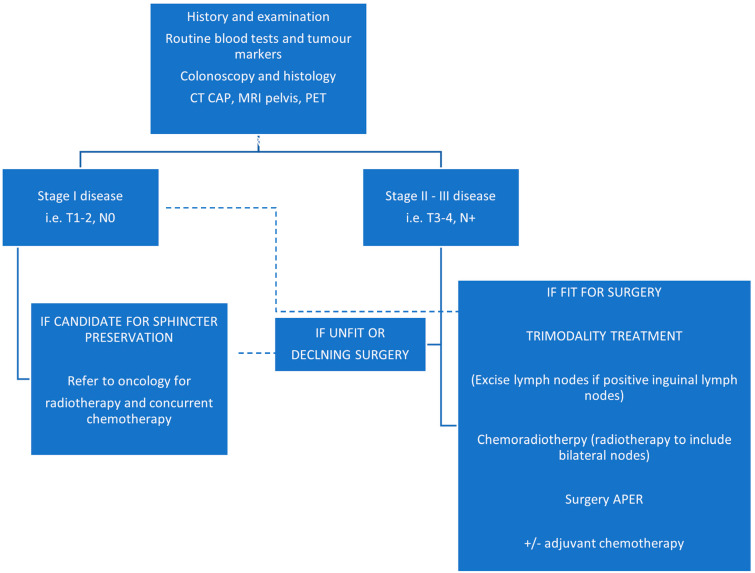
Overview of anal adenocarcinoma management (reproduced from Lukovic et al. [6]).

**Table 1 cancers-14-03738-t001:** Overall survival rates by treatment at 5 years (primary outcome measure).

5-Year Overall Survival by Treatment (%)						
		S Only	CRT + S	S + CRT	CRT Only	RT or CT Only
1	Peiffert 2012 [8]	-	-	-	75.0 (CFS)	-
2	Bertelson 2015 [9]	-	-	-	-	-
3	Franklin 2016 [10]	-	-	-	-	-
4	Su 2017 [11]	-	-	-	-	-
5	McKenna 2019 [12]	-	-	-	-	-
6	Leong 2019 [13]	-	-	-	-	-
7	Wang 2019 [14]	-	-	-	-	-
8	Lewis 2019 [15]	-	-	-	-	-
9	Li 2019 [16]	-	61.1	-	39.8	-
10	Malakhov 2019 [17]	57.6	64.6	51.7	39.2	-
11	Wegner 2019 [18]	69.1	64.1	67.3	42.0	-
12	Park 2020 [19]	77.7 (CSS)	80.3	65.8	63.9	35.7
13	Gogna 2020 [20]	-	-	-	-	-
14	Yasuhara 2021 [21]	-	-	-	-	-
15	Chatani 2021 [22]	-	-	-	-	-

**Table 2 cancers-14-03738-t002:** Important extracted data of included studies, including OS and recurrence (secondary outcome measures).

Study	N	N with Fistulas (F) or Crohn’s (CD)	Age (Median)	Intervention (s)	Median Follow-Up (Months)	Recurrence (N)	Median OS (Months)	OS (%)	Quality Scale (NOS)
Peiffert (2012) [8]	307	n/a	58.8 (mean)	CRT/ CRT + ICT	50	R = 88/307 LR = 68/88. DM =20/88	n/a	75% at 5 years (CRT) 76.5% at 5 years (CRT + ICT)	8
Bertelson (2015) [9]	18	F = 3, CD = 2	53 (mean)	S Only (APR)/ S (APR) + CRT/ CRT/ CT Only	25, 17, 8	LR = 4, DM = 10	24, 17	n/a	6
Franklin (2016) [10]	462	n/a	69	No treatments compared	n/a	n/a	33	30.2% at 5 years	7
Su (2017) [11]	126	n/a	55.5	S Only/ S + CRT/ CRT/ Palliative	30	LR = 36 DM = 25	n/a	43.4% at 5 years	7
McKenna (2019) [12]	2117	n/a	n/a	S/ CT/ RT/ Combinations	n/a	n/a	65	n/a	8
Leong (2019) [13]	5	5	64	CRT + S/ S + CRT/ CRT	n/a	LR = n/a, DM = 3/5	10.5	n/a	6
Wang (2019) [14]	136	n/a	60	S (APR) only/ S + RT/ S + CT/ S + CRT	44	n/a	n/a	n/a	8
Lewis (2019) [15]	1183	0	n/a	CRT/ CRT + S (APR)	150	n/a	72.5	55.9% at 5 years	6
Li (2019) [16]	1747	n/a	n/a	CRT Only/ CRT + S(APR)	41.1	n/a	n/a	61.1% at 5 years (CRT + S) 39.8% at 5 years (CRT only)	8
Malakhov (2019) [17]	1193	n/a	66	S Only/ CRT + S/ S + CRT/ CRT	47.6	n/a	n/a	48.4% at 5 years	7
Wegner (2019) [18]	1729	n/a	65	S Only/ CRT + S/ S + CRT/ CRT	55	n/a	69, 92, 83, 45	55% at 5 years	8
Park (2020) [19]	393	n/a	65	S Only/ CRT + S/ S + CRT/ CRT/ RT or CT	29	n/a	n/a	72.9% at 3 years (CSS)	6
Gogna (2020) [20]	2090	n/a	68.12	CRT/ CRT + S	n/a	n/a	n/a	39.6% at 5 years	6
Yasuhara (2021) [21]	102	CD = 34	56	S only/ S + CRT	54.9	LR = 26, DM = 25	n/a	91% at 5 years (CA) 85.7% at 5 years (NCA)	6
Chatani (2021) [22]	359	n/a	65, 62	S + CRT/ CRT + S (LE or APR)	n/a	n/a	85.8, 65.3	n/a	6

**Table 3 cancers-14-03738-t003:** Main conclusion from each study included in this systematic review.

	From This Study	Main Conclusions
1	Peiffert et al. (2012) [8]	No advantage for induction chemotherapy (ICT) or HD radiation boost use
2	Bertelson et al. (2015) [9]	For stage II AA patients CRT followed by APR is the treatment choice, with curative resection offering no significant long-term DFS outcomes
3	Franklin et al. (2016) [10]	Consider more aggressive therapy since AA has worse prognosis than SCCA and RA
4	Su et al. (2017) [11]	Prophylactic inguinal nodal treatment necessary for AA patients, even if negative ILNs
5	McKenna et al. (2019) [12]	Increased mortality associated with non-surgical management thus AA patients need MDT evaluation and surgery referral
6	Leong et al. (2019) [13]	Treatment of choice is multimodal with neoadjuvant CRT followed by APR (CRT + S)
7	Wang et al. (2019) [14]	AA has worse prognosis than RA and T staging criteria for anal carcinoma may not be valid for AA
8	Lewis et al. (2019) [15]	Trimodality therapy offers better survival outcomes than CRT alone, specifically CRT followed by APR within 6 months
9	Li et al. (2019) [16]	CRT followed by surgery (CRT + S) associated with significant OS benefit
10	Malakhov et al. (2019) [17]	AA tends to be treated like rectal cancer using neoadjuvant CRT and a more aggressive approach necessary with surgery, particularly APR, being important
11	Wegner et al. (2019) [18]	Improved OS by incorporating surgery in AA management compared to CRT alone
12	Park (2020) [19]	CRT given preoperatively with surgical resection might maximise OS outcomes
13	Gogna et al. (2020) [20]	Survival outcomes significantly improved with surgery
14	Yasuhara et al. (2021) [21]	Outcomes Crohn’s disease-associated patients with larger sized AA tumours are significantly poorer. Improved outcomes of CRT + S compared to S only.
15	Chatani et al. (2021) [22]	No overall survival difference between local excision or APR in combination with CRT

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
