# Peer review of "Management and Outcomes in Anal Canal Adenocarcinomas—A Systematic Review"

_cancers, 2022, doi:10.3390/cancers14153738_

Round 1

Reviewer 1 Report

Overall, this is a nice review of a rare GI cancer -- anal adenocarcinoma.  The rarity of the disease does mandate reviews of  the literature to try to address treatment patterns and outcomes.  The authors are commended for tackling this subject.

Specific issues to be addressed:

1.  The diagnostic dilemma of whether the cancer is a rectal adenocarcinoma v. anal adenocarcinoma was not addressed in the review.  The authors would improve on the manuscript by discussing this and how this may account for differences in the reported outcomes of the various studies.

2. Given the rarity of the disease and that this is a review, the manuscript can be improved with some basics on the incidence, demographics of the disease.  In line 64, the authors discussed the survival rates in England but do not give the annual numbers, incidence rates, male v. female ratio, etc.  

3. In line 66 - 67, the authors link anal adenocarcinoma to HPV.  More discussion on the etiology of anal adenocarcinoma since while Squamous cell carcinoma of the anal canal is more than 90% related to HPV, what isthe rate of HPV related anal adenocarcinoma?

4. Table 1 would be better off deleting the studies without survival data and limiting the list of stuides to 9 - 12.  In addition, the n of each study can be added as a column.  Alternatively, maybe Table 1 can be combined with Table 2.  

5. Minor issues: Please keep the initials consistent. In places APER was used for abdominoperineal resection (Line 321) v. APR (Lines 345, 359). 

Reviewer 2 Report

Excellent revision of rare topic.

The main drawback is the heterogeneity of the studies and its retrospective character.

Methodology correct but the language limitation could increase the sample size.

Author Response

Thank you for your comments. We agree that heterogeneity of the data and retrospectivity are the main the drawbacks of the subject matter.

Anal adenocarcinoma is a rare entity making it hard for a consensus to be reached with regards to its management, as it is not possible conduct large scale studies with sufficient number of patients.  Therefore, a retrospective approach had to be used to study the benefits of potential treatments.

In response to your comments, the writing author performed an informal search of literature that did include non english language papers - however this did not broaden the catchment as these papers had already been translated into english. 

I hope this answers your comments.

Reviewer 3 Report

The manuscript represents an interesting systematic review. 

The introductory part must be developed. The authors should described the treatment variants. 

Please explain why neoadjuvant chemoradiotherapy (CRT) followed by radical surgery of abdominoperineal excision of rectum (APER) is named trimodality treatment.  

Methods: the authors should precise the key words used for searching the literature. 

The authors should clearly specify the place of radiotherapy for these types of tumors, in general and in their study.

Round 2

Reviewer 3 Report

The value of the manuscript increased.